# New Viral Facets in Oral Diseases: The EBV Paradox

**DOI:** 10.3390/ijms20235861

**Published:** 2019-11-22

**Authors:** Lilit Tonoyan, Séverine Vincent-Bugnas, Charles-Vivien Olivieri, Alain Doglio

**Affiliations:** 1Faculté de Chirurgie Dentaire, Université Côte d’Azur, EA 7354 MICORALIS (Microbiologie Orale, Immunothérapie et Santé), 06357 Nice, France; Severine.VINCENT@univ-cotedazur.fr (S.V.-B.); charlesolivieri@sfr.fr (C.-V.O.); 2Pôle Odontologie, Centre Hospitalier Universitaire de Nice, 06001 Nice, France; 3Unité de Thérapie Cellulaire et Génique, Centre Hospitalier Universitaire de Nice, 06103 Nice, France

**Keywords:** Epstein–Barr virus, lichen planus oral, periodontal diseases, Sjogren’s syndrome, etiopathogenesis of oral inflammatory diseases, viral-bacterial synergism, plasma cells

## Abstract

The oral cavity contributes to overall health, psychosocial well-being and quality of human life. Oral inflammatory diseases represent a major global health problem with significant social and economic impact. The development of effective therapies, therefore, requires deeper insights into the etiopathogenesis of oral diseases. Epstein–Barr virus (EBV) infection results in a life-long persistence of the virus in the host and has been associated with numerous oral inflammatory diseases including oral lichen planus (OLP), periodontal disease and Sjogren’s syndrome (SS). There is considerable evidence that the EBV infection is a strong risk factor for the development and progression of these conditions, but is EBV a true pathogen? This long-standing EBV paradox yet needs to be solved. This review discusses novel viral aspects of the etiopathogenesis of non-tumorigenic diseases in the oral cavity, in particular, the contribution of EBV in OLP, periodontitis and SS, the tropism of EBV infection, the major players involved in the etiopathogenic mechanisms and emerging contribution of EBV-pathogenic bacteria bidirectional interaction. It also proposes the involvement of EBV-infected plasma cells in the development and progression of oral inflammatory diseases. A new direction for preventing and treating these conditions may focus on controlling pathogenic EBV with anti-herpetic drugs.

## 1. Introduction

With all due respect to microbial flora, the mouth is home to the second most diverse microbial community in the human body, housing a variety of microbes, including bacteria, viruses and fungi [1]. The oral cavity is also the main portal of entry for pathogens into the human host. The mouth is not a homogeneous environment for the resident microbes but constantly changing distinct bio-habitats, such as teeth, gingival sulcus, attached gingiva, tongue, cheek, lip, hard and soft palate, which are occupied by different microbial communities. The oral microbial community is normally in equilibrium and lives in symbiosis with a healthy host. The transition in the microflora towards disease development happens as a result of a complex interaction of microbial-specific traits, host immune responses and ecosystem-based factors [2,3]. The bacteria and fungi have been traditionally the hub in the discussions of microbiological aspects of oral diseases. The bacteria and fungi responsible for the most common oral diseases are now better known, while viruses have a more questionable reputation. Viruses, though, are probably more involved in oral diseases than has been previously recognized [4]. Indeed, the oral mucosal surfaces are also associated with a complex viral population (virobiota). Our current understanding of how these viruses replicate and persist within the oral tissues is still in infancy and better knowledge of the oral virobiota represents an emerging facet of modern oral biology. The analysis of viral diversity has been notably afforded by the advances in sequencing and bioinformatics technologies improving our knowledge of oral virobiota. Many of the viruses identified so far in the human virome are bacteriophages, opening a way to new exciting discoveries about the role of bacteriophages in oral microbiology [5]. In addition, major eukaryotic viral families occurring in the oral cavity and contributing to the propagation of oral diseases in adult human individuals include herpesviruses, papillomaviruses, picornaviruses/enteroviruses, retroviruses [6,7,8,9] and the recently identified redondoviridae [10].

The present review will be particularly focused on Epstein–Barr virus (EBV), a herpesvirus showing a typical oral ecology and commonly found in most inflammatory oral lesions. This viral infection, usually considered as a benign infection in most healthy individuals, is however associated with various malignancies, functional abnormalities of immunity and oral diseases [11,12,13]. Whether EBV is only an opportunistic agent which benefits from inflammatory conditions to actively replicate, or a critical pathogen able to trigger and worsen oral diseases remains an intriguing unsolved question. This review aims to bring new light to this paradox.

## 2. EBV Biology

Herpesvirus species are one of the most prevalent viral families in the oral cavity. Herpesviruses have coexisted with the human host over millions of years, developing sophisticated strategies to stay in their host for its lifetime. Following a primary infection, herpesviruses manifest a latent infection in a small number of specific cellular sites, minimizing viral production. This allows them to evade the immune system and persist in the host with minimal impact on them [14]. Acute infection and periodic reactivation, on the other side, allows the infectious virus to spread to new hosts [13,15,16].

EBV belongs to this ancient and highly successful herpesvirus family. It is often said that EBV infection is one of the most successful chronic viral infections in humans. Indeed, EBV is strikingly ubiquitous in humans, establishing a persistent, almost benign infection in over 90% of the world’s adult population for their entire life [13]. Asymptomatic primary infection with EBV typically occurs during infancy and early childhood. In contrast, delayed primary infection is contracted in young adults who have not been exposed to EBV early in life. Symptomatic acute infection at this time in 25–80% of cases manifests as infectious mononucleosis (IM) [15,17], dubbed the “kissing disease”. Most often, EBV is transmitted from the carrier to naïve host orally through direct contact with virus-infected saliva, such as with kissing, also via blood and semen during sexual contact, blood transfusion and organ transplantation [18]. EBV can infect a variety of cell types under different circumstances. It preferentially colonizes B lymphocytes, but can also infect epithelial cells (EC), occasionally, T lymphocytes, natural killers (NK), monocytes, smooth muscle cells and, possibly, also follicular dendritic cells (DC) [19].

There has been a controversy over which cell type is infected first by EBV following salivary transmission, being either oropharyngeal ECs or directly B cells. It has been speculated that EBV actually first infects the oropharyngeal ECs, where it replicates, and then is released to infect B cells co-localized in the lymphoepithelial structures [15] (Figure 1). An alternative model suggests, that EBV enters into the crypts of lymphoepithelium of the tonsils via saliva, crosses the epithelial barrier (which is often only one cell thick, immediately above the bed of lymphocytes) and directly infects naïve B cells [20]. In agreement with this last model, Dunmire and co-authors [21] showed that EBV viral genomes are detected at low levels in blood about three weeks before any IM symptoms, suggesting early systemic diffusion of EBV-infected B cells before epithelial amplification in the oral cavity.

EBV entry into the target cells is the fundamental part of the viral infectivity and happens as a result of the fusion of viral and host membranes mediated by viral glycoproteins and cellular receptors. Our understanding of EBV entry into the different cell types is incomplete, but some of the major players involved in B cell and EC infections have been identified [19]. EBV envelope glycoprotein gp350/220 has a major role in the attachment of EBV to the complement receptor type 2 or cluster of differentiation 21 (CR2/CD21), a B cell-specific surface receptor. The internalization of EBV requires also the interaction of another viral glycoprotein gp42 with human leukocyte antigen (HLA) class II molecules on B cells. On the other hand, the attachment of EBV to ECs is a more complicated event and is less understood. In ECs lacking CD21 and HLA class II, two possible attachment mechanisms have been proposed: (1) entry via gHgLR alone—two viral glycoproteins gH and gL directly bind to gHgLR receptors on ECs; (2) entry via integrins and gHgLR—EBV transmembrane envelope glycoprotein BMRF2 interacts with integrins on ECs (β1 family and α5β1) possibly initiating signaling events, while gHgL interacts with gHgLR. Previous studies indicated that the integrins αvβ5, αvβ6 and αvβ8 function as gHgLR receptors for EC entry [22,23]. However, more recent studies have identified the ephrin receptor A2 (EphA2) as a critical player for EBV EC entry [24,25]. It was notably shown in these studies that EphA2 specifically bound to gHgL and antibodies against EphA2, or EphA2 inhibition, blocked EC infection. Although EBV uses different mechanisms to attach to different cell types, the fusion of EBV with either B cells or ECs occurs by a common fusion machinery consisting of the fusion “regulator”, gHgL complex, and the ‘‘executor’’ fusion protein, gB.

EBV drives B cell infection, differentiation, persistent infection, reactivation and reinfection (Figure 1), mimicking antigen activation by switching patterns (programs) of different sets of viral gene expression at key points along the way [27]. Infected B cells in vivo may express four different programs of gene usage. One program is used to produce infectious virions (lytic cycle) in antibody-producing cells, namely plasma cells (PC). The lytic cycle is characterized by the sequential expression of immediate early genes (IE), followed by early genes (E) and late genes (L), a cascade regulated by specific viral transactivators, in particular, BZLF1 and BRLF1 [28,29]. The other three programs are all associated with latent infection and long-term EBV persistence in the body: the growth program (latency 3), when all nine known viral latent proteins are expressed (six Epstein–Barr virus nuclear antigens: EBNAl, EBNA2, EBNA3A–C) and EBNA leader protein (EBNA-LP) and three latent membrane proteins: LMP1, LMP2A–B, including non-coding RNAs (EBV-encoded small RNA: EBERs); the default program (latency 2), when a restricted set of three latent proteins are expressed (EBNA1, LMP1 and LMP2A); EBNA1 only program (latency 1) and latency program (latency 0) when none of the latent genes are expressed [13]. Upon infection of the tonsilar naïve B cells, EBV drives the infected cells to become proliferating B cell blasts using the growth program. EBV-infected blasts migrate into the lymphoid follicle, where they switch to the default program and initiate a germinal center (GC) reaction. EBV-infected GC B cells leave the follicle and exit to the periphery as resting memory B cells which are quiescent as the virally encoded gene expression is completely silenced, allowing the infection to persist without detection by the immune system (latency program). These cells in the periphery occasionally undergo proliferation, which is not driven by EBV but by the normal memory homeostatic mechanism. In this dividing memory B cells, EBV expresses the genome tethering protein EBNA1 which allows the viral genome to replicate with the cells (EBNA1 only program). Latently infected memory B cells continuously circulate in the periphery and return to the lymphoid tissue, where they differentiate into PCs in response to antigen stimulations or unknown stimuli and release infectious virions (lytic program) to initiate a new round of infection of naïve B cells. This periodic reactivation in the tonsils leads to shedding of EBV into the oral cavity and transmission both to a new host and to previously uninfected B cells within the same host, thus completing the cycle (Figure 1). The detailed discussion of these events is presented in reviews published by Thorley-Lawson and colleagues [13,20,30].

It looks as if EBV has evolved successful strategies to minimize its pathogenic potential, for the sake of persisting the infection and the survival of the host in which it persists. It follows, therefore, that the virus should only create a pathogenic threat and contribute to the development of EBV-associated diseases when something goes astray.

## 3. Immune Response to EBV

To address the EBV infection, the host has to maintain efficient immune functions, including innate and adaptive immunities [12]. The innate immune system provides the first line of defense against the EBV infection. Neutrophils, monocytes/macrophages, DCs and specifically NKs are important in controlling early EBV infection until the adaptive immune system establishes long-lasting immune control [31,32]. NKs rapidly recognize and directly eliminate newly-infected B cells (Figure 1). They can also produce interferon (IFN) γ and interleukin (IL) 2 to augment NK cell and antigen-specific T cell responses, thereby limiting the EBV infection [33]. The adaptive immune response to EBV infection includes both humoral and cellular immune responses. Antibodies limit the spread of the infectious virus in the late phases of infection, while cytotoxic T cells destroy infected cells that express viral antigens and control both the primary and persistent phases of EBV propagation [30,32]. EBV stimulates strong humoral responses and production of antibodies, such as immunoglobulin (Ig) G, IgM and IgA, which recognize various EBV targets, including EBV structural antigens (e.g., glycoproteins gp350/220), lytic antigens (e.g., some of the IE and E lytic cycle proteins) and latent antigens (e.g., EBNA1 and LMPs). Cytotoxic T cells are the major determinants in the cellular control of acute EBV infection and are directed against both lytic and latent antigens [32]. Both CD4^+^ helper and CD8^+^ cytotoxic T cells make a robust response to EBV antigens (Figure 1), however, the CD8^+^ T cell response is the most dramatic reaction to EBV challenge. Primary EBV infection (as seen during IM) induces exaggerated response and over-proliferation of virus-specific CD8^+^ T cells (with lytic-antigen reactivity dominant over latent), though smaller bursts of virus-specific CD4^+^ T cell reactivity can also be detected [12]. In any case, the immune system is unable to eliminate the virus completely and in a minute fraction of B cells, the virus acquires a latent state with minimal immune exposure and persists for life [30].

Evidently, both the virus and the host have evolved multiple elaborate strategies to prevent the possible fatal consequences of EBV infection. However, if the delicate balance between the host immune system and the virus is not maintained, it may lead to EBV-associated diseases. This review discusses novel viral aspects of the etiopathogenesis of non-tumorigenic diseases in the oral cavity, in particular, the contribution of EBV in oral inflammatory diseases, the tropism of EBV infection and the major players involved in the etiopathogenic mechanisms.

For the literature search, we identified the most relevant articles reporting the above-mentioned topics using web-based search engines including Google, Google Scholar and PubMed database. Searches were performed using different combinations of free text words and medical subject heading (MeSH) terms. Reference lists of selected articles were hand-searched for other relevant publications.

## 4. The Relationship between EBV and Oral Inflammatory Diseases

### 4.1. Oral Lichen Planus

Oral lichen planus (OLP) is a chronic inflammatory disease of the oral mucosa with malignant potential [34]. OLP affects 0.5–1% of the population worldwide, generally is a disease of the middle-aged and elderly, and the female-to-male ratio is about 1.4:1 [35]. OLP is characterized by lesions consisting of white, grey, velvety, thread-like papules in a linear, annular and retiform arrangement forming lacy, reticular patches, rings and streaks [36]. The lesions typically occur on the buccal mucosa, the dorsum of the tongue and the gingiva, but rarely on the palate or lip [37]. There are different clinical manifestations of OLP including reticular, papular, plaque-like, erosive, atrophic or bullous types. Burning sensation and pain usually accompany the erosive, atrophic and bullous type lesions. The main complication of OLP is the reduced quality of life, related to soreness or pain. OLP may also have associations with systemic diseases, but most importantly, the risk of malignant change has been recognized [37].

OLP, as virtually all diseases, result from the interplay of host, lifestyle and environmental factors. The exact etiopathogenesis of OLP has been a subject of much research. Several factors have been implicated in the etiology and several hypotheses have been proposed for its pathogenesis. The various mechanisms involved in the pathogenesis include antigen-specific cell-mediated immune response, non-specific mechanisms, humoral immunity and autoimmune response (reviewed more detailed in [36,37,38,39,40]). The popular hypothesis in the literature describes OLP as a T cell-mediated inflammatory, autoimmune-like disease in which cytotoxic CD8^+^ T cells trigger the apoptosis of basal keratinocytes in the oral epithelium. An early event in the disease induction presumably includes a putative endogenous or exogenous agent(s) binding to keratinocytes. Keratinocytes are thought to express or unmask the OLP antigen. Following this, T cells (mostly CD8^+^, and some CD4^+^ cells) migrate into the epithelium either for the routine surveillance and encounter the antigen by chance (“chance encounter” hypothesis) or via the attraction of keratinocyte-derived chemokines (“directed migration”). Migrated CD8^+^ cells are activated directly by the antigen binding to the major histocompatibility complex (MHC) I on a keratinocyte or through activated CD4^+^ cells. Activated CD8^+^ cells, in turn, kill basal keratinocytes through tumor necrosis factor (TNF) α (when T cell-secreted TNFα binds the TNFα R1 receptor on the keratinocyte surface), Fas-FasL-mediated (when T cell surface CD95L (Fas ligand) binds the CD95 (Fas) on the keratinocyte surface) or granzyme B-activated (T cell-secreted granzyme B entering the keratinocyte via perforin-induced membrane pores) mechanisms, all which activate a caspase cascade resulting in keratinocyte apoptosis.

The antigen responsible for inducing OLP is still unidentified. Although several genetic, environmental, physiological and infectious factors have been proposed for the etiology, which include genetic background, dental materials, drugs, infectious agents (bacterial and viral infections), autoimmunity (associated with other autoimmune diseases), immunodeficiency, food allergies, stress, habits, trauma, diabetes and hypertension, malignant neoplasms and bowel disease [36,37,38,41]. One interesting association of OLP etiology is the EBV infection.

There are several indications that EBV can play a role in the etiopathogenesis of OLP: (i) EBV is a common oral pathogen with high salivary and gingival occurrence; (ii) EBV is frequently found in oral inflammatory lesions, including autoimmune lesions; (iii) EBV is considered to be involved in the malignant transformation of epithelial cells, and OLP is considered to have malignant potential; (iv) CD8^+^ cytotoxic T cells are known to trigger apoptosis of virally infected cells, hence, EBV infection may be involved in the pathogenesis of OLP. The correlation between EBV and OLP has been the focus of many studies, although any causal role remains speculative and controversial. Several techniques for EBV detection have been used by these studies, among them polymerase chain reaction (PCR) or nested PCR, immunohistochemistry (IHC) and in situ hybridization (ISH).

Early studies reported that keratinocyte expression of the EBV receptor (CD21) was up-regulated in OLP lesions [42] and humoral responses to EBV were altered in OLP patients [34]. Using the PCR technique, Cruz and colleagues [43] reported that 50% of the OLP cases were positive for the EBV DNA; Sand and co-authors [44] detected significant prevalence of the EBV DNA in OLP patients (26.1%) as compared to healthy subjects (7.3%); Kis and the group [45] showed that patients with OLP carried significantly more EBV DNA than controls (46.6% and 19.1%, respectively); Shariati and colleagues [46] reported statistically significant occurrence (15.8%) of the EBV DNA in the OLP group. Whereas, Yildirim et al. [47] immunohistologically detected the statistically significant prevalence of EBV in 35% of OLP cases. In contrast to these studies, recently Danielsson and co-authors [48] did not detect evidence of EBER1 and EBER2 expression by EBER-ISH (ISH to detect small noncoding EBV-encoded RNAs) in the biopsies of OLP patients and concluded that EBV was not involved in the etiology of OLP. Though the majority of the published literature showed the occurrence of EBV in OLP lesions, it is yet unclear whether EBV is involved in the etiopathogenesis of OLP or it is secondary to a local or general decrease in immune defense.

A recent comprehensive study [49] was carried out on a large series of clinically representative OLPs using EBER-ISH. They detected a significant presence of EBV in OLP lesions (74%), and more strikingly in the severe erosive form of OLP (83%) as compared to mild reticular form (58%). The EBV-positive cells represented the large part of the infiltrate and were significantly correlated with local inflammatory parameters, suggesting a direct relationship between EBV infection and inflammatory status. Moreover, OLP lesions were mostly composed of CD138^+^ PCs rather than CD20^+^ B cells. On the other hand, the electron microscopy of the EBV^+^ PCs showed that EBV was actively producing viral particles, suggesting possible amplification of EBV infection within the lesion. Of note here, PCs, though, have a major antibody-producing function [50], a regulatory role for these cells is increasingly discussed [51,52]. There were also profound changes in cytokine expression of EBV^+^ PCs, notably, increased expression of major pro-inflammatory products (i.e., CXCL1, IL1β, CXCL2 and IL8). This study [49] brings conclusive evidence showing that OLP is commonly infiltrated with EBV^+^ PCs, allowing a first glimpse inside the “black box” of OLP’s etiopathogenesis in relation to EBV.

### 4.2. Periodontal Disease

The periodontal disease is regarded as one of the commonest oral diseases and the most common chronic inflammatory disease in humans [53]. Periodontal disease refers to a set of pathological chronic inflammatory conditions that affect the structures surrounding and supporting the teeth (the gingiva, periodontal ligament and alveolar bone), which could lead to tooth mobility and eventual tooth loss and contribute to systemic inflammation [54]. The periodontal disease initiates as gingivitis (inflammation of the gingiva), which is highly widespread and readily reversible by effective oral hygiene. When left untreated, it may progress to chronic periodontitis (CP), a more serious condition (mostly irreversible) that destroys tooth-supporting connective tissues and alveolar bone. The World Health Organization (WHO) estimates that in about 20% of middle-aged (35–44 years) world population the disease can progress further to irreversible severe periodontitis that may rapidly result in tooth loss [55]. Poor oral health has significant consequences on health, notably, tooth mobility being related to pain, discomfort, halitosis, and tooth loss affecting food intake, nutrition, self-esteem and premature aging. Beyond oral health, the periodontal disease may also be associated with several systemic severe diseases such as diabetes, cardiovascular diseases, adverse pregnancy outcomes and respiratory diseases [56,57]. Moreover, a recent prospective large cohort study provides additional evidence that cancer risk, especially for lung and colorectal cancer, is elevated in individuals with periodontitis [58].

The etiopathogenesis of periodontal disease is commonly linked to specific bacteria and viruses, protective and destructive host immune responses, genetic and epigenetic factors, and modifiable and non-modifiable environmental factors [59]. Traditionally, the pathogenesis of periodontitis was mainly explained as the infection caused by pathogenic bacteria that colonize tooth surface and gingival sulcus. Extensive research efforts have been dedicated to the study of the periodontal disease-associated microflora, from classic culture-dependent methods to modern approaches on the molecular, whole genomic and proteomic levels. Up to 800 different species have been identified in human dental plaque so far [54]. The debate on which species are particularly pathogenic and initiate disease has lasted decades and is not solved yet. The most frequently identified periodontopathic bacteria include microaerophilic species such as *Actinobacillus actinomycetemcomitans*, *Campylobacter rectus* and *Eikenella corrodens*, and, especially, anaerobic species *Porphyromonas gingivalis*, *Bacteroides forsythus*, *Treponema denticola*, *Prevotella intermedia*, *Fusobacterium nucleatum*, *Eubacterium* and spirochetes [56].

Although bacteria are necessary for the infectious periodontal disease to arise, it is now evident that the pathogenesis occurs when the balance between the oral microbial biofilm and the host is lost, owing to dysbiosis and immune overreaction of the host to the microbial presence [54]. Pathogenic bacteria colonizing tooth surface and gingival sulcus initiate a local inflammatory reaction that activates the immune system. The innate immune system provides immediate defense against infection and inflammation by (i) recruiting immune cells such as neutrophils, monocytes and activated macrophages; (ii) activation of the complement system and release of chemical mediators such as cytokines (TNF, ILs, IFNγ and transforming growth factor (TGF) β); (iii) activation of the adaptive immune system that represents more specific response to injury or inflammation (proliferation of antigen-specific T and B cells with consequent antibody production, assisting macrophages and generating killer cells). This hyper-inflammatory response leads to alveolar bone resorption by osteoclasts, and degradation of ligament fibers attaching the tooth to the bone by matrix metalloproteinases, and the situation becomes largely irreversible [60].

Though the significance of bacteria in the etiopathogenesis of all types of periodontal diseases is undeniable, bacterial-host interaction alone does not adequately explain various clinical characteristics of the disease. In his review, Slots [2] pointed out those features: (i) episodic progressive nature; (ii) localized pattern of tissue destruction; (iii) periodontitis of many subjects affects relatively few teeth despite the ubiquitousness of periodontopathic bacteria in saliva; (iv) alveolar bone destruction of certain teeth while adjacent teeth remain untouched; additionally, (v) surface treatments to eliminate dental plaque can only stabilize the disease, but do not result in the cure; (vi) reappearance of the disease occurs after antibiotic therapy; (vii) antiviral treatments significantly improve the clinical outcomes of the periodontal condition [61,62]. These uncertainties and observations have stimulated efforts to search for additional etiopathogenic factors for periodontitis.

During the past decade, herpesviruses have emerged as putative pathogens in various types of periodontal diseases [63]. In particular, human cytomegalovirus and EBV have been closely associated with severe types of periodontitis. Amongst the many arguments for a herpesvirus involvement in periodontal disease, Slots, in his review [63], signified the following observations: (i) genomes of the two herpesviruses occur abundantly in different types of progressive periodontitis of children, adolescents and adults; (ii) presence of herpesviruses in periodontal sites increase with disease severity; (iii) periodontal inflammatory cells contain nucleic acid sequences of herpesviruses; (iv) herpesvirus-positive periodontitis lesions also harbor increased levels of periodontopathic bacteria. Based on the observations on herpesvirus-bacteria-host interactive responses, a new paradigm was proposed in periodontitis pathogenesis involving a viral and bacterial combination to promote long-term chronic inflammatory disease: the herpesviral-bacterial synergy hypothesis [61,62,64].

There is extensive literature demonstrating and favoring the link between EBV and periodontal diseases (reviewed in [2,59,61]). The majority of these studies evidenced that EBV can be detected with high abundance in periodontal lesions, that it directly infects gingival ECs and that the viral load positively correlates with disease severity. In addition, a recent comprehensive meta-analysis has summarized the associations of EBV and periodontal diseases based on existing literature, concluding that high frequent detection of EBV correlates with increased risk of periodontal diseases [65]. Furthermore, there is also a good deal of evidence suggesting a positive relationship between periodontitis, EBV, as well as periodontopathic bacteria, and reviews authored by Slots describe in more detail the interaction between periodontal herpesviruses and periodontopathic bacteria [64,66].

The periodontopathic role of EBV is, thus, suggested from the association studies, but the specific underlying mechanisms by which EBV may cause or progress periodontitis remain unclear. In the context of a proposed periodontopathic bacteria-EBV synergistic model, in this review, we have chosen to present the most interesting findings of 3 studies [67,68,69] which we believe shed much light on the contribution of each pathogen in the etiopathogenesis of periodontitis.

Vincent-Bugnas and colleagues [69] investigated EBV infection in healthy and CP gingival samples to show that EBV infection was widespread not only in CP samples but also detectable in healthy specimens before the onset of periodontitis. Moreover, they made a surprising discovery that EBV was present in the periodontal epithelium (pECs) that surrounds and attaches teeth to the gingiva and that the level of EBV-infection in pECs correlated with disease severity. These suggest that the oral pool of EBV-infected ECs in healthy individuals may have implications for viral shedding. Also, this study demonstrated that EBV-infected pECs were more prone to be apoptotic than non-infected pECs. As a consequence, EBV may stimulate a pro-inflammatory response in infected cells, notably by stimulating the CCL20 (chemokine C-C motif ligand 20) production, a pro-inflammatory chemokine known to regulate the Th17/Treg axis (pro-inflammatory T helper 17/suppressive regulatory T cell balance) [70]. In turn, the presence of apoptotic pECs may compromise the integrity of the gingival epithelium, promoting bacterial colonization by increasing the adherence to ECs and/or by allowing a release of “danger signals” by injured cells to favor biofilm development.

Bacteria, on the other hand, may reactivate latent EBV. It was proposed that a novel mechanism of epigenetic regulation, such as histone modification, plays an important role in the maintenance and disruption of viral latency [71]. In particular, hypoacetylation of histone proteins in the BZLF1 (IE gene encodes the master transcription factor ZEBRA controlling the transition from latency to lytic replication) promoter by histone deacetylases (HDACs) is essential for the maintenance of EBV latency. The researchers examined the effect of *P. gingivalis* infection on EBV reactivation and discovered that *P. gingivalis* induced expression of ZEBRA. They showed that culture supernatant from *P. gingivalis*, containing a significant amount of butyric acid, inhibited HDACs, thus increasing histone acetylation and the transcriptional activity of the BZLF1 gene in latently infected cells [67]. Similarly, Makino and group [68] investigated whether *P. endodontalis* can reactivate latent EBV by examining the induction of BZLF1 expression in vitro. Together, these studies suggest that the butyric acid produced by *P. endodontalis* or *P. gingivalis* may activate latent EBV in periodontal lesions, while the reactivated EBV can induce chronic inflammation and contribute to the clinical progression of periodontitis. Nevertheless, despite a large body of compelling research data, the conclusive etiopathogenic role of EBV in periodontitis development and progression is still to be explored.

### 4.3. Sjogren’s Syndrome

Sjogren’s syndrome (SS) is a chronic systemic autoimmune disease characterized by lymphocytic infiltration of salivary and lacrimal glands and epithelia resulting in glandular dysfunction and sicca symptoms—oral dryness (xerostomia) and ocular dryness (xerophthalmia) [72]. This condition can occur independently (known as a primary SS (pSS)) or jointly with underlying autoimmune disorders such as rheumatoid arthritis, systemic lupus erythematosus or progressive systemic sclerosis (secondary SS (sSS)). SS is a relatively common disease with a community prevalence of pSS ranging from 0.1–0.6% of the population [73]. pSS predominantly affects women with a female-to-male ratio of 9:1, with the mean age of onset usually in the 40–50 years of life [74]. Clinical manifestations of SS can be very broad, ranging from classic sicca symptoms, mild constitutional symptoms (such as fatigue, malaise, arthralgia) to systemic symptoms (such as neuropathy or vasculitis) [75]. Oral dryness can seriously affect the quality of life, hindering basic functions such as eating, speaking and sleeping. Reduction of salivary volume and subsequent loss of the antibacterial properties of saliva may accelerate infection, tooth decay and periodontal disease [76].

Despite the extensive studies, the underlying etiology of SS and its pathogenesis remain obscure. The etiopathogenesis of SS is thought to be a multifactorial interacting process: environmental (such as viral infections) or endogenous (hormones) factors trigger successive activation of innate and adaptive immune responses in a genetically susceptible host. Although a detailed discussion on the mechanisms of the pathogenesis of SS is beyond the scope of this article (reviewed elsewhere [77,78,79,80]), a proposed set of pathogenic events involves (i) initial insult to the gland that leads to cellular necrosis or apoptosis with subsequent expression of the SS-related antigen A (SS-A) on the glandular cell surface; (ii) activation of HLA-independent innate immune system that may respond to apoptotic products such as SS-A contained particles; (iii) liberation of chemokines and up-regulation of adhesive molecules that direct lymphocyte (predominantly CD4^+^ T cells) and DC migration into the gland; (iv) activation of T and B cells within the gland as a result of their interaction with ECs that express HLA-DR^+^, cell adhesion molecules and other co-stimulatory factors; (v) production of anti-SS-A antibodies by B cells under the influence of CD4^+^ T cells; (vi) secretion of pro-inflammatory cytokines by both lymphocytes and ECs to perpetuate the inflammatory response and production of IFN1 by the DCs, which further perpetuate the glandular migration of lymphocytes, activation of lymphocytes and metalloproteinases and apoptosis of glandular cells. Together, these inter-related processes trigger persistent immune system responses leading to tissue damage.

Among several environmental factors, the viral infection is one of the most likely candidates for the induction of SS. The possible candidate viruses include EBV, cytomegalovirus, human herpesvirus type 8, human T-lymphotropic virus type 1, hepatitis C virus and enterovirus [72].

Direct and indirect evidence in support of EBV involvement in the pathogenesis of SS includes the following: (i) EBV displays strong tropism for salivary gland (also at some extent for lacrimal gland) where it establishes a life-long infection in normal individuals, thus, the virus is present (in very low amounts) at the target organ of the immune response prior to the development of SS; (ii) EBV can stimulate a strong T and B cell responses, thus, the recognition of the virus at its site of latency or activation by the immune system may contribute to glandular tissue damage and eventual clinical symptoms of dryness; (iii) EBV viral load and EBV-directed antibodies can be found in the saliva, salivary biopsies and blood of SS patients in amounts greater than found in normal individuals. Additionally, (iv) SS patients are known to have an increased risk of development of EBV-associated lymphomas.

However, the efforts to link EBV and SS have resulted in controversial reports. Despite the common evidence of the persistence of EBV in the salivary glands, some studies observed no correlation between EBV infection and SS. Venables et al. [81], using ISH, detected the EBV DNA in ECs of salivary gland biopsy specimens from only 17% SS patients as opposed to 60% of healthy subjects, suggesting that EBV infection load was not increased in SS. The same group [82], using enzyme-linked immunosorbent assay (ELISA), found no evidence of elevated levels of anti-EBV antibodies in all SS patients. Similarly, DiGiuseppe and the co-authors [83] did not detect EBER1 in SS patients by ISH, suggesting that the latent EBV infection did not play a major pathogenic role in the lymphoepithelial salivary gland lesions associated with SS. On the other hand, several other studies, applying different techniques or combinations (PCR, IHC, ISH, ELISA, dot plot hybridization, immunoblotting), observed increased EBV nucleic acid and protein expression from sera, saliva, salivary and lacrimal gland biopsy specimens of patients with SS in comparison to healthy individuals, identifying mostly ECs and/or infiltrated B cells, as well as circulating B cells, as target sites for EBV infection [84,85,86,87,88,89,90,91,92,93,94].

To gain insight into the pathogenesis of pSS, interestingly, Song et al. [95] performed a microarray meta-analysis of publically available genome-wide gene expression disease studies to identify the most important genes and biological pathways associated with pSS. They observed that the majority of up-regulated genes mediated immune response, inflammation, chemotaxis and the interferon pathway. Moreover, the most significant pathway in their analysis was EBV infection.

The etiopathogenic link between EBV and SS can be further appreciated from a more recent study by Croia and colleagues [96], which suggests the answer might lie in ectopic lymphoid structures (ELS) within the salivary glands of SS patients, serving a unique home for latency and reactivation of EBV and driving the differentiation of PCs. ELS are defined as aggregates of lymphoid cells forming ectopically in nonlymphoid locations, which mimic many characteristics of B and T cell follicles allowing for a GC-like reaction to occur [97,98]. None of the previous studies appreciated the presence of these ectopic GC-like structures, though the association of EBV and GC reaction is extremely relevant, as EBV uses a GC-like growth program to gain access to the memory B cell compartment. The above-mentioned study [96], using a comprehensive array of techniques [IHC, immunofluorescence (IF), EBER-ISH, quantitative real-time reverse transcription PCR (RT-PCR) analysis], examined whether the expression of latent and lytic EBV occurs preferentially in the GC-like formations within salivary glands of SS patients and in situ differentiation of autoreactive PCs with specificity toward SS-associated autoantigens. They showed that (i) EBV LMP2A latent protein was selectively expressed by CD20^+^ B cells within ectopic follicles; (ii) expression of EBER (which is expressed during EBV latency) was strictly dependent on the formation of ELS in SS salivary glands; (iii) EBER was co-localized with CD20^+^ B cells and perifollicular CD138^+^ PCs; (iv) expression of the EBV BFRF1 protein (involved in lytic reactivation) was observed exclusively in CD138^+^ PCs, only in ELS^+^ SS salivary gland background. Thus, on one side ELSs provide a B cell compartment for EBV latent persistence and on the other side, they allow the reactivation of EBV giving rise to locally differentiated PCs. Most importantly, the authors also demonstrated that: (v) CD138^+^/BFRF1^+^ lytically infected PCs in SS were reactive with Ro52 (major autoantigen in SS) and that the autoreactive profile of EBV-infected PCs was disease-specific; (vi) these results were confirmed by the in vivo engraftment of ELS-containing SS salivary gland tissue into the severe combined immunodeficiency (SCID) mice, supporting the local production of antibodies against EBV and disease-specific autoantigens; lastly, (vii) cytotoxic CD8^+^ T cells accumulated outside B cell follicles, in areas rich in BFRF1^+^ cells, impairing CD8^+^ T cell-mediated cytotoxicity. The results from this study are indeed compelling to support the complex role for EBV in the SS pathogenesis. Although there is no conclusive evidence regarding the causative role of EBV in SS and the link remains to be further scrutinized, taken together these studies provide an intriguing possibility that EBV is a major active player in the etiopathogenesis of SS.

## 5. Proposed Mechanism of Involvement of EBV-Infected Plasma Cells in the Development and Progression of Oral Inflammatory Diseases

Taking into account the observations from several studies, we propose that the presence of EBV^+^ PCs in oral lesions may represent an exclusive unifying model supporting the pathogenic participation of EBV in the development and progression of OLP, CP and SS (Figure 2). Indeed, EBV^+^ PCs have been detected at high levels in OLP [49] and SS [96] and shown to be associated with the progression of these diseases. Regarding the CP, the presence of PCs in periodontal lesions is a hallmark of periodontitis [99,100] and current work from our team is also in favor of the presence of EBV-infected PCs in periodontitis (submitted manuscript).

First, PCs host the late stages of EBV replication and ensure the production of new virions. Cells harboring the lytic phase of EBV replication are strongly immunogenic and able to promote immune activation, notably, T-cell activation [13]. This antigenic stimulation can increase the inflammatory condition and the cross-talk with activated CD4^+^ T cells which, in turn, stimulates new memory B cell differentiation into antibody-producing PCs [101]. Furthermore, it was shown that several viral epitopes are shared with human proteins (molecular mimicry) leading to the proposal that increased production of antibodies against viral proteins may be highly pathogenic because of autoimmune reactivity [102]. Under normal conditions, a naïve B cell can only mature into a memory B cell after the surface HLA-D-bound peptides are recognized by the CD4^+^ T cell’s receptor. However, when infected with EBV, naïve B cells can mature into memory B cells without requiring a cross-talk with CD4^+^ T cells through the expression of EBV’s LMP1 and LMP2A genes. EBV infection can thus result in the emergence of memory B cell’s receptor recognizing normally “forbidden” epitopes to favor the proliferation of autoreactive B cells and the production of autoantibodies [101,103] which may worsen local inflammation.

Second, it was shown that EBV^+^ PCs grouped together to form high-density EBV^+^ cell clusters scattered within the infiltrate in close contact with neighboring tissues [49,96]. A working hypothesis is that the infiltration of EBV^+^ PCs and the massive production of EBV virions in the inflammatory region may favor extensive infection of the ECs, as well as B cells, promoting a self-amplifying process of EBV infection within the inflammatory lesions. In that context, infection and destruction of epithelia by EBV are very likely, taking into account that EBV entry in polarized ECs is favored by close contact with EBV-producing PCs [104]. These infected oral epithelial tissues, in turn, may represent a reservoir for further intense EBV replication and EBV shedding in saliva and spreading to distant sites.

Last, in addition to their role in antibody production, growing evidence has demonstrated that PCs also behave as major regulators of immune functions [51,52]. PCs can produce anti-inflammatory cytokines IL10 and IL35, suggesting immune suppression since these cytokines are extensively implicated in tolerogenic mechanisms. Interestingly, a recent study showed that the feature of periodontitis-associated PCs was to produce the IL35/IL37 anti-inflammatory cytokines [99]. Consequently, the anti-inflammatory activity induced by EBV^+^ PCs could alter the vigilance of host immune players and promote the infection of bacterial pathogens. Bacteria, on the other hand, can produce compounds reactivating latent EBV in the lesions (i.e., *P. endodontalis* and *P. gingivalis* producing butyric acid triggering the transition of latent EBV to lytic [67,71]). This viral-bacterial synergy, proposed as a vicious circle concept [64], promotes the long-term chronic inflammatory disease. On the other side, regulatory PCs may also produce pro-inflammatory cytokines and chemokines which augment the inflammation and, if not contained, have an enhanced risk of tissue destruction. As an illustration, the study from our group indicates that OLP lesions, heavily infiltrated with EBV^+^ PCs, were associated with increased expression of major pro-inflammatory products i.e., CXCL1, IL1β, CXCL2 and IL8 [49]. Of particular interest, IL1β is one of the most important mediators of the inflammatory response and IL8 is considered as the primary cytokine involved in the recruitment of neutrophils to the site of damage or infection.

## 6. Conclusions

Undoubtedly, oral inflammatory diseases represent serious global health problems with significant social, economic and health-system impacts. The development of effective therapies, therefore, requires better and deeper insights into the etiology and pathology of these diseases. The etiopathogenic role of EBV cannot be underemphasized as it has shown to modify disease conditions. Although EBV stands out as a consistent and prominent pathogenic factor, the EBV paradox still exists, as a critical aspect of the host immune response, the important role of other identified or unidentified bacteria, as well as an emerging contribution of EBV-bacteria bidirectional interaction cannot be excluded.

EBV, as well as other herpesviruses, may exert oral pathogenic potential through different mechanisms, operating alone or in combination with bacteria. This interaction between herpesviruses and bacteria may be bidirectional, with bacterial enzymes or other inflammation-inducing products activating herpesviruses (vicious circle) [64]. Various synergies between virus, bacteria and host are only just becoming understood, and clearly indicate that inter-kingdom interactions play a major role in shaping health or disease. In any case, the concept that EBV, and other oral herpesviruses, play an important role in oral inflammatory diseases has therapeutic implications, and a new direction for preventing and treating these conditions may focus upon controlling pathogenic herpesviruses with specific anti-herpetic drugs [62].

## Figures and Tables

**Figure 1 ijms-20-05861-f001:**
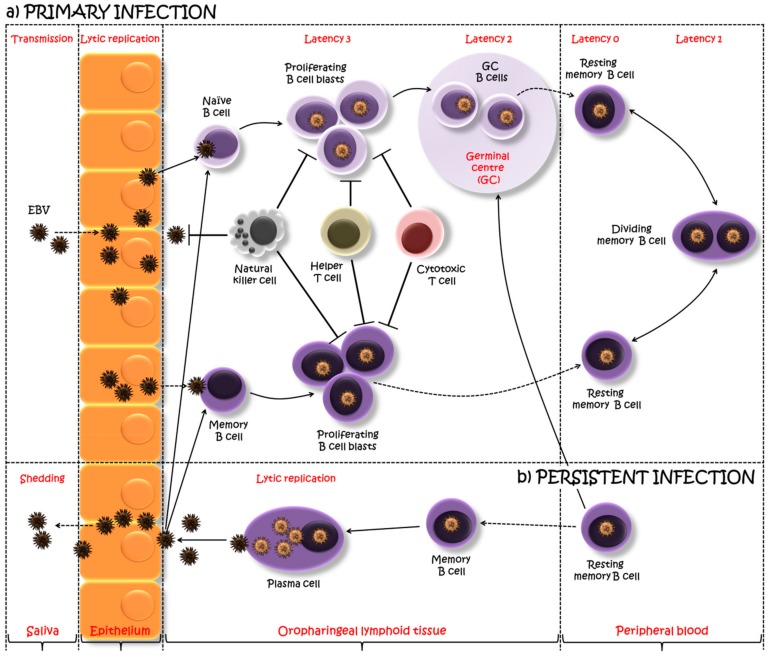
A model of Epstein–Barr virus (EBV) primary and persistent infection. EBV, transmitted via saliva, lytically replicates, possibly in the oropharyngeal epithelium, after which the virus spreads throughout the lymphoid tissues and infects underlying B cells. One viewpoint proposes that EBV infects tonsilar naïve B cells turning them into latently infected proliferating B cell blasts, using the growth program or latency 3, and expressing a full spectrum of latent proteins. The majority of these proliferating cells are removed by natural killer and T cell response. However, the surviving B cell blasts undergo a germinal center (GC) reaction, where a more limited set of viral genes are expressed (the default program or latency 2). The viral persistence is realized through the silent infection of B cells when EBV-infected GC B cells migrate to peripheral blood establishing a stable reservoir of resting memory B cells, in which no EBV gene is expressed (latency program or latency 0). Intermittent expression of EBNA1 during the division of these memory B cells allows the viral genome to be distributed to the daughter memory B cells (EBNA1 only program or latency 1). An alternative viewpoint suggests infection of pre-existing memory B cells as a direct route into the memory B cell reservoir. Occasionally, these EBV-infected cells might be recruited into GC reactions, after which they might either re-enter the reservoir as memory B cells or return to the lymphoid tissue and undergo plasma cell differentiation, activating the viral lytic cycle. This allows the EBV replication, shedding into saliva and transmission both to new hosts and to previously uninfected naïve B cells within the same host. Adapted from [13,26].

**Figure 2 ijms-20-05861-f002:**
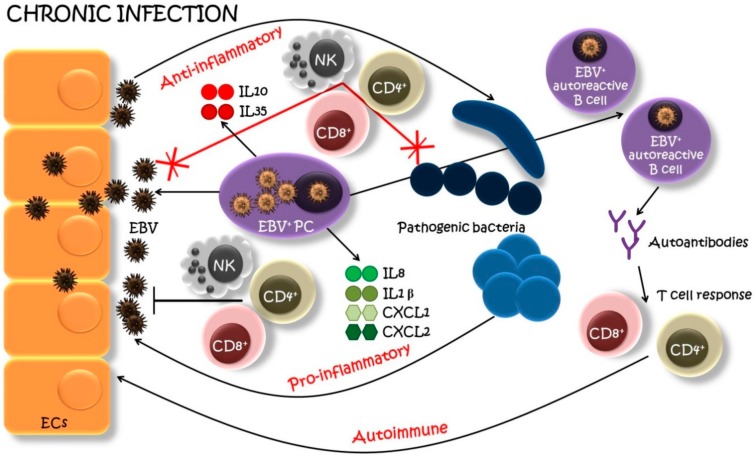
A model of EBV-infected plasma cell’s (PC) involvement in chronic oral infections. EBV-infected PCs (EBV^+^ PCs) infiltrate the site of the oral infection and actively produce EBV within a lesion. New viral particles infect neighboring epithelial cells (EC), which serve as an EBV pool. EBV^+^ PCs, aside of producing immunoglobulins, secret also pro-inflammatory cytokines and chemokines (IL1β, IL8, CXCL1, CXCL2), recruiting host immune players (natural killers (NK), helper T cells (CD4^+^), cytotoxic T cells (CD8^+^), etc.) to generate a pro-inflammatory reaction. Furthermore, the EBV infection can result in the emergence and proliferation of autoreactive B cells with “forbidden” epitopes favoring the production of autoantibodies and recruitment of T cells specific for lytic-cycle viral antigens with a further autoimmune attack. On the other side, EBV^+^ PCs may also produce anti-inflammatory cytokines (IL10, IL35) suppressing the host immune surveillance which leads to the superinfection of pathogenic bacteria. These bacteria, in turn, may reactivate the latent EBV and the production of new viral particles. And the circle starts anew leading to the progression of the infection [49,69,96,101,105].

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
