# Peer review of "New Viral Facets in Oral Diseases: The EBV Paradox"

_ijms, 2019, doi:10.3390/ijms20235861_

Round 1

Reviewer 1 Report

This review article is about Epstein-Barr virus (EBV) and its involvement in certain oral inflammatory diseases (oral lichen planus, periodontal disease and Sjorgen’s syndrome). The topic is interesting and the text is generally well put together. However, it is unclear how the cited literature for this review was selected, whether a systematic approach (i.e. a search algorithm) was employed, whether the cited references represent a comprehensive or a selective collection of publications in this field, and what inclusion or exclusion criteria were for the cited publications. The authors should add a brief Method section to clarify how the literature was compiled for this review and which databases were screened.

Although the text is understandable, there are many grammatical flaws, which should be corrected. Careful proofreading, preferentially by a native English speaker, is recommended. Here are some suggestions for corrections (but there are many more mistakes):      

Line 65: is one of the most successful chronic viral infections …

Line 205: The popular hypothesis in the literature describes OLP as a T cell-mediated …

Line 244: Please, explain “ISH”.

Line 300: activation of the complement system …

Line 344: in the etiopathogenesis of periodontitis …

Line 360: plays an important role

Line 439: can be further appreciated

Line 447: In the current study [96], Unclear. Do you mean “in the recent study”?

Fig. 2: The black text on the dark backgrounds is difficult to decipher. Please optimize.

Author Response

Point 1: This review article is about Epstein-Barr virus (EBV) and its involvement in certain oral inflammatory diseases (oral lichen planus, periodontal disease and Sjorgen’s syndrome). The topic is interesting and the text is generally well put together. However, it is unclear how the cited literature for this review was selected, whether a systematic approach (i.e. a search algorithm) was employed, whether the cited references represent a comprehensive or a selective collection of publications in this field, and what inclusion or exclusion criteria were for the cited publications. The authors should add a brief Method section to clarify how the literature was compiled for this review and which databases were screened.

Response 1: We thank the reviewer for the careful and thorough reading of this manuscript. We hope that the reviewer will find our responses satisfactory.

Please note that this review is a narrative and not a systematic review. Here we give a comprehensive overview of main topics and discuss the most relevant studies.

We added a paragraph to clarify how the literature was compiled and which databases were screened.

L191-194: “For the literature search, we identified the most relevant articles reporting the above-mentioned topics using web-based search engines Google, Google Scholar and PubMed database. Searches were performed using different combinations of free text words and medical subject heading (MeSH) terms. Reference lists of selected articles were hand-searched for other relevant publications.”

Point 2: Although the text is understandable, there are many grammatical flaws, which should be corrected. Careful proofreading, preferentially by a native English speaker, is recommended.

Response 2: We have carefully revised the manuscript to improve the English. Corrections have been made in the text as track changes.

Point 3: Here are some suggestions for corrections (but there are many more mistakes):     

Line 65: is one of the most successful chronic viral infections …;

Line 205: The popular hypothesis in the literature describes OLP as a T cell-mediated …

Response 3: The proposed corrections have been made.

Point 4: Line 244: Please, explain “ISH”.

Response 4: ISH was already explained in L234 of the first version of the manuscript (now L242-243): “…immunohistochemistry (IHC) and in situ hybridization (ISH).

Point 5: Line 300: activation of the complement system …;

Line 344: in the etiopathogenesis of periodontitis …;

Line 360: plays an important role;

Line 439: can be further appreciated.

Response 5: Corrected as requested.

Point 6: Line 447: In the current study [96], Unclear. Do you mean “in the recent study”?

Response 6: The former sentence:

“In the current study [96], the authors, using a comprehensive array of techniques [IHC, immunofluorescence (IF), EBER-ISH, quantitative real-time reverse transcription PCR (RT-PCR) analysis], investigated whether the expression of latent and lytic EBV occurs preferentially in the GC-like formations within salivary glands of SS patients and in situ differentiation of autoreactive PCs with specificity toward SS-associated autoantigens.”

now reads as:

“The above-mentioned study [96], using a comprehensive array of techniques [IHC, immunofluorescence (IF), EBER-ISH, quantitative real-time reverse transcription PCR (RT-PCR) analysis], examined whether the expression of latent and lytic EBV occurs preferentially in the GC-like formations within salivary glands of SS patients and in situ differentiation of autoreactive PCs with specificity toward SS-associated autoantigens.”

Point 7: Fig. 2: The black text on the dark backgrounds is difficult to decipher. Please optimize.

Response 7: Thank you for the suggestion. The figure is optimized now.

Reviewer 2 Report

Dear Authors,

Thank you very much for submitting the work to International Journal of Molecular Sciences. I have read the manuscript carefully and the reviewer thinks that improvements must be made for the present from. The reviewer has written the comments in the below. Thank you very much.

(1)What is novelty of this study?

(2)What is clinical relevance?

(3)Line 23 It seems that it may not be mandatory but it may be better to use MESH keyword.

https://www.ncbi.nlm.nih.gov/mesh 

(4)Are there any additional diseases that may be affected by EBV? 

(5)Have the authors performed research regarding EBV and oral lichen planus or EBV and periodontal disease? 

The reviewer would like to suggest the authors to submit to another journal focused on viral diseases.

Thank you very much.

Author Response

Point 1: Dear Authors,

Thank you very much for submitting the work to International Journal of Molecular Sciences. I have read the manuscript carefully and the reviewer thinks that improvements must be made for the present from. The reviewer has written the comments in the below. Thank you very much.

Response 1: We thank the reviewer for reviewing our manuscript.

Point 2: What is novelty of this study?

Response 2: In this review, we discuss the involvement of EBV in the etiopathogenesis of oral non-tumorigenic inflammatory diseases, in particular, oral lichen planus, periodontitis and Sjogren’s syndrome. Previous reviews in the literature mostly discuss the association of EBV with malignancies of the oral cavity rather than with inflammatory lesions. This discussion is given in L187-479.

Here, we also review the studies which examined the bacteria-EBV synergistic model in the etiopathogenesis of periodontitis. This point was discussed in L349-383.

Besides, we propose the involvement of EBV-infected plasma cells in the development and progression of oral inflammatory diseases. As it was pointedly marked by Reviewer 3, “In such a case, it opens doors for exploring the therapeutic efficacy of anti-viral drugs in treatment of these inflammatory conditions”. This point was stated in L480-542.

Point 3: What is clinical relevance?

Response 3: This review points to the role of EBV in the development of oral inflammatory diseases and proposes to focus on the control of pathogenic EBV with antiviral drugs for the prevention and treatment of these conditions. This is suggested in L20-22, L557-560.

Point 4: Line 23 It seems that it may not be mandatory but it may be better to use MESH keyword.

https://www.ncbi.nlm.nih.gov/mesh

Response 4: Thank you for this suggestion. Keywords found in MeSH are replaced with MeSH terms, such as lichen planus, oral; periodontal diseases; Sjogren’s syndrome; plasma cells.

Point 5: Are there any additional diseases that may be affected by EBV?

Response 5: EBV infection has been linked to the development of various lymphoid and epithelial human malignancies, such as Burkitt’s lymphoma, Hodgkin lymphoma, nasopharyngeal carcinoma, gastric carcinoma, as well as multiple autoimmune diseases like rheumatoid arthritis, systemic lupus erythematosus and multiple sclerosis.

Point 6: Have the authors performed research regarding EBV and oral lichen planus or EBV and periodontal disease?

Response 6: Yes, the authors published a research article regarding EBV and oral lichen planus in the Journal of Dental Research (reference 49 in the manuscript – Raybaud H, Olivieri CV, Lupi-Pegurier L, Pagnotta S, Marsault R, Cardot-Leccia N, Doglio A. Epstein-Barr virus-infected plasma cells infiltrate erosive oral lichen planus).

Regarding EBV and periodontal disease, the authors have one published research article in Plos One (reference 69 in the manuscript – Vincent-Bugnas S, Vitale S, Mouline CC, Khaali W, Charbit Y, Mahler P, Prêcheur I, Hofman P, Maryanski JL, Doglio A. EBV infection is common in gingival epithelial cells of the periodontium and worsens during chronic periodontitis) and one submitted manuscript.

Point 7: The reviewer would like to suggest the authors to submit to another journal focused on viral diseases.

Thank you very much.

Response 7: We think that this manuscript aligns well with the topic of the special issue “Etiopathogenesis of Virus Associated Oral Diseases" from the section "Molecular Pathology, Diagnostics, and Therapeutics".

Reviewer 3 Report

In the manuscript titled “New viral facets in oral diseases: the EBV paradox”, the authors have performed an intense review of the role of EBV in various oral inflammatory conditions like oral lichen panus, periodontitis and Sjogren syndrome. They have also proposed a mechanism for the role of EBV+ plasma cells in the development and progression of oral inflammatory diseases. In such a case, it opens doors for exploring the therapeutic efficacy of anti-viral drugs in treatment of these inflammatory conditions.

The authors should correct “Sjorgen” to “Sjogren” in the manuscript What is the authors opinion about using anti-viral medications for treatment of oral lichen planus cases which are EBV+? If such cases of OLP were cured with anti-viral medications, would that change the description of OLP which is now considered a persistent condition? Did the authors find any studies of EBV+ lichen planus cases which were treated with anti-herpetic drugs?

Author Response

Point 1: In the manuscript titled “New viral facets in oral diseases: the EBV paradox”, the authors have performed an intense review of the role of EBV in various oral inflammatory conditions like oral lichen panus, periodontitis and Sjogren syndrome. They have also proposed a mechanism for the role of EBV+ plasma cells in the development and progression of oral inflammatory diseases. In such a case, it opens doors for exploring the therapeutic efficacy of anti-viral drugs in treatment of these inflammatory conditions.

Response 1: We would like to thank the reviewer for spending time on our behalf and reviewing our manuscript. We thank the reviewer to state that we “have performed an intense review”, “proposed a mechanism for the role of EBV+ plasma cells in the development and progression of oral inflammatory diseases” and that our proposal “opens doors for exploring the therapeutic efficacy of anti-viral drugs in treatment of these inflammatory conditions”.

Point 2: The authors should correct “Sjorgen” to “Sjogren” in the manuscript. 

Response 2: Thank you for this correction. The corrections have been made.

Point 3: What is the authors opinion about using anti-viral medications for treatment of oral lichen planus cases which are EBV+?

Response 3: The authors think that topical or systemic antiviral drug therapy may be indeed favourable in such cases. Considering the antiviral treatment is rather affordable and available, it has a chance to be readily implemented in clinical practice.

Point 4: If such cases of OLP were cured with anti-viral medications, would that change the description of OLP which is now considered a persistent condition?

Response 4: We think that antiviral medications may be of therapeutic value in the future. However, before surely talking about how antiviral drugs change the course of OLP progression or achieve complete eradication of this disease randomized controlled clinical trials in appropriate patient groups are obviously needed.

Point 5: Did the authors find any studies of EBV+ lichen planus cases which were treated with anti-herpetic drugs?

Response 5: We found studies that suggested treating EBV-positive OLP patients with anti-viral drugs (such as Adtani and Malathi, 2015; Shariati et al, 2018). However, in our literature search within Google, Google Scholar, PubMed databases we could not find any case studies which actually performed the antiviral treatment.

Round 2

Reviewer 1 Report

Thank you for the amendments. The manuscript is much better now..

Author Response

Comment: Thank you for the amendments. The manuscript is much better now.

Response: We would like to thank the reviewer for positive comments, constructive suggestions and efforts towards improving our manuscript.

Reviewer 2 Report

Dear authors.

Thank you very much for preparing the revised manuscript. The reviewer was happy to review the valuable manuscript. The reviewer thinks that improvements have been made.

Thank you very much.

Author Response

Comment: Dear authors.

Thank you very much for preparing the revised manuscript. The reviewer was happy to review the valuable manuscript. The reviewer thinks that improvements have been made.

Thank you very much.

Response: We would like to thank the reviewer for the work in revising our paper and for the positive comments.